# Overweight and obesity among Vietnamese school-aged children: National prevalence estimates based on the World Health Organization and International Obesity Task Force definition

**Huong Duong Phan[1], Thi Ngoc Phuong Nguyen[2], Phuong Linh Bui[2,3]\*, Thanh Tung Pham[2,4], Tuan Vu Doan[1], Duc Thanh Nguyen[1], Hoang Van Minh[2]**

**1** National Hospital of Endocrinology, Hanoi, Vietnam, **2** Center for Population Health Sciences, Hanoi University of Public Health, Hanoi, Vietnam, **3** Department of Nutrition, Harvard T.H.Chan School of Public Health, Boston, MA, United States of America, **4** Department of Epidemiology, Harvard T.H.Chan School of Public Health, Boston, MA, United States of America

\* bpl@huph.edu.vn

**Data Availability Statement:** There is an ethical restriction on sharing a de-identified dataset due to the ethical requirement of local IRB, namely

## Abstract

### Background

Overweight and obesity is a severe global health issue in both developed and developing nations. This study aims to estimate the national prevalence of overweight and obesity among school-aged children in Vietnam.

### Method

We conducted a national cross-sectional study on 2788 children aged from 11–14 years old from September to November 2018. We applied the WHO 2007 and IOTF criteria to estimate the prevalence of overweight and obesity among participants. Poison regression analysis with cluster sampling adjustment was employed to assess associated factors with obesity and overweight. Metadata on sociodemographic characteristics, physical measurements, and lifestyle behaviors were also extracted to investigate these factors in association with overweight and obesity prevalence.

### Results

The prevalences of overweight and obesity in Vietnamese children were 17.4% and 8.6%, respectively by WHO Z-score criteria, and 17.1% and 5.4%, according to the IOTF reference. Using WHO Z-score yielded a higher prevalence of obesity than the IOTF and CDC criteria of all ages and both sexes. The proportions of overweight and obesity were substantially higher among boys than girls across ages. Parental BMI was shown to be a significant factor associated with overweight/obesity status in both girls and boys. Only for boys, age (PR = 0.83, 95% CI 0.76–0.90) and belonging to ethnic minorities (PR = 0.43, 95% CI 0.24–0.76) were significant risk factors for overweight/obesity.

Institutional Science Review Board of the National Hospital of Endocrinology. If anyone interested in our data, they may directly contact our research team via email (ntnp@huph.edu.vn) or the IRB member (PhD. MD. Le Quang Toan, Vice Chairman of the IRB of National Hospital of Endocrinology via letoan.endo@gmail.com or (+84) 904007097).

**Funding:** This study was supported by the State budget for prevention of hazardous infectious diseases and common non-communicable diseases (Funding number: 0649).

**Competing interests:** The authors have declared that no competing interests exist.

## Conclusion

Our findings indicate a high prevalence of childhood overweight and obesity in Vietnam, especially in boys.

## Introduction

The physical state of being overweight or obese are two of the most severe global health challenges [1]. According to a report from the World Health Organization in 2016, over 2 billion people suffer from these conditions, of which over 340 million are children and adolescents [1]. It is estimated that obesity accounts for approximately 2.8 million deaths and 36 million disability-adjusted life years (DALYs) (2.3% of the total global DALYs) annually [2]. Among all risk factors for adult obesity, excessive weight gain in childhood and adolescence accounts for the most significant risk factors [1]. Unfortunately, the burden of overweight and obesity among children and adolescents is increasingly notable in Asia [3–5], particularly among low and middle-income countries like Vietnam [6–9].

To diagnose childhood obesity, two of the most widely used methods in the literature are the World Health Organization (WHO) 2007 [10] and the International Obesity Task Force (IOTF) [11] guidelines, both of which employ Body Mass Index (BMI) as the primary indicator. However, since the WHO methods is primarily based off of collected US population data, whereas IOTF made use of the multinational database, the overweight/obesity prevalence is slightly diverse estimated when using different due to the use of different classification methods. In 2010, the Vietnamese Ministry of Health reported that the national prevalence of obesity and overweight among children (aged 6 to 19 years) was 8.5% and 0.7%, respectively [12]. However, this report only provides information using the WHO Z-score criteria and does not provide information in regards to the sampling method [12]. There have been some studies which examine the obesity prevalence among Vietnamese children and adolescents currently [13–20]. However, a major limitation of these studies is that the investigators used either WHO Z-score or IOTF reference for calculating and surveyed small districts of Vietnamese metropolis (mostly in Ho Chi Minh city) [13, 14, 16–20].

Apart from assessing the prevalence, it is important to explore the diversity of factors which contribute to the onset of childhood obesity; this is necessary to establish an effective and long-term public health strategy for the prevention of this morbidity. For example, a higher socio-economic status [4, 13–15, 21], increase in sedentary lifestyles [4, 13, 15], unbalanced dietary habits [4, 13, 15, 22], and parents' overweight/obesity status [4, 13, 21, 23] are known contributors to an increased risk of childhood obesity in many countries. Whereas, the evidence of such association among Vietnamese children is quite limited. All current studies on childhood obesity in Vietnam have indicated an association between overweight/ obesity risk and sexes [13–17, 19, 20]. Specifically, only three of these studies identified higher risk among children who live in urban areas, with wealthier families, are physical inactive, live with parental obesity, and have an unhealthy diet [13–15]. However, as mentioned previously, these surveys are conducted with miniscule sample size, in small areas and use different criteria for obesity diagnosis. Thus, the conclusions derived from these studies could not generalize that childhood obesity prevalence is representative of the whole population.

Collectively, an assessment of obesity prevalence among Vietnamese children using a national representative sample and comparing multiple diagnostic criteria is necessary. Such evidence would become the foundation to develop effective childhood overweight/obesity

prevention programs in Vietnam. In this study, we aim to estimate the national prevalence of overweight and obesity in children by using both the WHO and IOTF criteria and to identify associated factors which influence these conditions among school-aged children in Vietnam.

## Method

### Study design and setting

We employed a cross-sectional study design on a national population-based sample. Sampling was performed from September to November of 2018, the study involved all three regions of Vietnam (North, Center, and South). This study was led by the National Hospital of Endocrinology in collaboration with the Department of Health and the Department of Education and Training in each province. Ethical approval for this study was obtained from the Institutional Science Review Board of the National Hospital of Endocrinology (No 1133/QD-BVNTTW).

### Inclusion criteria

Students were recruited into the study if they met the following inclusion criteria: (1) aged 11 to 14 years at the time of the survey; (2) free from physical or mental illness; (3) received approval to participate from their parents; and (4) willingness to participate in the study.

### Sample size and sampling method

We calculated the necessary sample size for our project using a cross-sectional study formula [24] with p = 8.5%, this criteria is based on a previous national report detailing children aged 11–13 years who were at risk for obesity [12]. Additionally, we multiplied this sample size with the design effect (DE) of 2. An increase of 10% of the sample size was further added for cases in which some participants would refuse to participate or be absent during the collection period. Our final sample size, thus, was rounded up to a total of 2880 children.

In regard to the sampling method, we employed a multistage cluster sampling method to select children aged 11–14 years in 3 main regions of Vietnam (North, Center, and South). The probability proportionate to size (PPS) method was utilized to select for clusters, which were defined as either wards, communes or towns. The flow chart of our sampling method was presented in S1 Fig.

At first, a list of demographic clusters and their respective populations were obtained from the 2009 Vietnam Population and Housing census [25]. In each region, 30 clusters were randomly selected, and a total of 90 clusters which represented 55 (out of total 64) provinces of Vietnam were included. Within each cluster, only one secondary school was chosen as a sampling unit. If the cluster had more than one secondary schools, the school with the highest number of students would be selected. If the selected cluster was without a secondary school, we selected another school within the previously mentioned criteria within the nearest random neighbor ward/commune/town. After schools were identified, we obtained a list of all classes in four cohorts (grades sixth to ninth), randomly selected one class in each cohort for each school, respectively. For every selected class, eight students (four girls and four boys) were chosen at random. If any student failed to meet any of the previously described inclusion criteria, we would select another student of the same sex within the selected class. In the end, 32 students within each cluster were included in the study.

### Survey instruments and physical measurement

After obtaining the written approval from the participants' parents, a team of health professionals who were trained as data collectors by the National Hospital of Endocrinology,

National Institute of Nutrition, and local endocrinology hospitals in the provinces were assembled. After explaining and obtaining verbal agreement from the children, this team collaboratively interviewed all students using a paper-based questionnaire and took their physical measurements. This questionnaire was designed based on the previous studies in Vietnam among adult population and the doctoral thesis of Tran et al on children aged 6–14 in Hanoi, Vietnam [26, 27]. The questionnaire was not validated in Vietnamese children but pretested for grammatical errors and readability among 30 children in Ha Long city, Quang Ninh province, Vietnam before the full-scale data collection. All the questions were revised and finalized by a panel of health experts from National Hospital of Endocrinology and Hanoi University of Public Health. During the period of data collection, all students were given the right to withdraw from the study at any stage. It was noted to the students that their withdrawal would not affect their relationship with the school, teachers, health care providers, and their school performance.

Our paper-based questionnaire included a variety of questions on several socioeconomic factors (including age, ethnicity, region, resident type), and general health status (such as physical activities and lifestyle behavior) (S1 File). Age of participants was derived from self-reported date of birth and date of anthropometric. For both the participants' height and weight, measured participants worn only light-weight clothing without shoes during measurements. Height was measured in centimeters (cm) and rounded to the nearest 0.1 cm by use of the Microtoise stature meter, weight was measured in kilograms (kg) and rounded to the nearest 0.1 kg with a Tanita digital scale (model BC-543, Tanita Corporation, USA). We further directly collected information about children's body fat percentage though this scale. Parents' weight and height were collected via an independent survey sent directly to them for self-report (S1 File). Body Mass Index (BMI) of parents was then calculated as the individual's body mass (kg) divided by the square of their height ($m^2$).

## Criteria for childhood overweight/obesity

Our main outcomes of interest were overweight and obesity in a population of Vietnamese children. We defined overweight and obesity by using the sex- and age-specific BMI ($kg/m^2$) cut off-points, as recommended by the WHO 2007 Z-score [10] and the International Obesity Task Force (IOTF) 2000 reference population [11]. These two criteria have been recommended for obesity diagnosis/screening [28–30] and are widely applied in a variety of countries, such as India, China, South Korea, Portugal, Iran, Canada, Turkey, and Italy [23, 31–40].

In regards to the WHO Z-score as applied to both for children and adolescents within the age of 5–19 years, "overweight" is defined as a BMI of 1 SD (standard deviations) to below 2 SD above the normal mean, whereas a BMI of 2 SD and above is considered as "obesity" [10]. Meanwhile, the IOTF criteria also used age- and sex-specific BMI values, but these cut-off points for defining "overweight" and "obesity" are defined to pass through BMI of 25 and 30 $kg/m^2$ at the age of 18, respectively [11]. The detailed IOTF cut off points for each sex and age group is described elsewhere [11].

Besides the two aforementioned criteria, we also used the CDC 2000 cut-off points to calculate the prevalence of overweight and obesity [41]. This cut-off points utilized in the CDC 2000 are 85th and 95th percentile, respectively [41]. For the parents of our participants, we used the BMI cut-off points of 23–25 $kg/m^2$ and $\geq 25$ $kg/m^2$ to categorize them as overweight or obese, respectively according to WHO recommendation for Asian adults population [42].

Further, we also utilize the cut-off point of below (-2SD) height-for-age and weight-for-height to calculate the prevalence of stunting and thinness (wasting) [43, 44].

## Statistical methods

A descriptive analysis stratified by sex was applied. BMI-for-age score was calculated using the WHO 2007 Stata package. This package was described in detail and published elsewhere [45]. The mean, standard definition (SD), median, and interquartile interval (IQI) were calculated separately for each sex. Several tests such as the Pearson's chi-squared tests, two-sample t-test, Wilcoxon rank-sum test, one-way ANOVA test, and the Kruskal-Wallis test were employed to examine the differences between sexes and BMI groups. In this study, the sex ratio of 1 [male]: 1 [female] was the most similar to the most recent National Housing Census funded by the Vietnamese government in 2009 for the same age group (the sex ratio of 1.04 [male]: 1 [female]) [25]. Additionally, our sampling method and sampling collection also reflected this equal ratio. Therefore, for calculating the total prevalence of both overweight and obesity, we did not adjust for oversampling by sex. The Stata *Survey* package was used to adjust for cluster sampling and to calculate the prevalence of overweight and obesity in our sampled population. Due to the high prevalence of overweight and obesity observed (larger than 10%) in this study, the use of logistic regression models could potentially lead to an overestimation of the associations between these disease outcomes and potential risk factors [46]. Therefore, to assess these associations more accurately, we employed a Poison regression analysis with cluster sampling adjustment for the calculation of Prevalence Ratios (PRs) [46]. All analyses were conducted with Stata 14.1 (StataCorp, College Station, TX, USA)

## Results

There were 2788 children (96.81%) with completed information on age, sex, weight and height included in the analysis. Among the 2880 interviewed participants, 70 records (2.43%) were found to have incomplete information on age (41 cases), sex (11 cases), weight (12 cases), and height (10 cases).

**Table 1** illustrates the general characteristics, physical measurements, and lifestyle behaviors of the participants in the present study. As a result of our sampling method, we investigated an equal number of male and female students, in which the study population was distributed evenly among age groups and regions. The vast majority of the participants were descended from the Kinh ethnicity (84.1%), and most of participants resided in rural areas (59.0%). The mean BMI in both male and female children was assessed to be 19.44 (±4.30) and 19.00 (±3.60), respectively. Additionally, we identified that the height, weight, and BMI-for-age of male participants was slightly higher than the indexes of their female counterparts ($p < 0.001$). However, in this study population we found that the body fat percentage was significantly higher in girls as compared to boys ($p < 0.001$). We also observed that the interviewee's fathers had a higher BMI as compared to their mothers (mean: 22.73 kg/m$^2$ vs. 21.95 kg/m$^2$).

Bicycles were reported as the most common means of transportation for students to go to school. Almost all students played at least one type of sport, the mean time per day for physical exertion was approximately 60 minutes for boys and 45 minutes for girls ($p < 0.001$). Most of the boys who participated in this study reported playing soccer/football, while girls mostly played badminton and shuttlecock kicking. Boys spent significantly more time playing video games, as compared to girls ($p < 0.001$). Regardless of sex, approximately one in three children spent more than 2 hours watching TV or using a smartphone daily.

The prevalence of overweight and obese according to different diagnostic criteria is presented in **Table 2**. Overall, 17.9% (95%CI: 15.1–19.9) of participants were found to be overweight according to the WHO Z-score reference. Similarly, the IOTF overweight prevalence was assessed to be 17.1% (95%CI: 14.7–19.9). However, the obesity prevalence differed vastly

**Table 1. Characteristics of all participants.**

| | Boys (n = 1398) | Girls (n = 1390) | Total (n = 2788) | p-value |
|---|---|---|---|---|
| **Demographic characteristics** | | | | |
| Age group | 1398 (100) | 1390 (100) | 2788 (100) | |
| 11 years old, n (%) | 362 (25.9) | 352 (25.3) | 714 (25.6) | 0.91 [a] |
| 12 years old, n (%) | 348 (24.9) | 359 (25.8) | 707 (25.4) | |
| 13 years old, n (%) | 371 (26.5) | 358 (25.8) | 729 (26.1) | |
| 14 years old, n (%) | 317 (22.7) | 321 (23.1) | 638 (22.9) | |
| Ethnicity | 1396 (100) | 1388 (100) | 2784 (100) | |
| Kinh, n (%) | 1174 (84.1) | 1167 (84.1) | 2341 (84.1) | 0.99 [a] |
| Other, n (%) | 222 (15.9) | 221 (15.9) | 443 (15.9) | |
| Region | 1398 (100) | 1390 (100) | 2788 (100) | |
| North, n (%) | 464 (33.2) | 463 (33.3) | 927 (33.2) | 0.82 [a] |
| Center, n (%) | 459 (32.8) | 469 (33.7) | 928 (33.3) | |
| South, n (%) | 475 (34.0) | 458 (32.9) | 933 (33.5) | |
| Residency status | 1398 (100) | 1390 (100) | 2788 (100) | |
| Urban, n (%) | 581 (41.6) | 561 (40.4) | 1142 (41.0) | 0.52 [a] |
| Rural, n (%) | 817 (58.4) | 829 (59.6) | 1646 (59.0) | |
| **Physical measurement** | | | | |
| Height | 1398 (100) | 1390 (100) | 2788 (100) | |
| Mean (SD) | 153.62 (10.62) | 150.67 (7.49) | 152.15 (9.31) | **<0.001 [b]** |
| Median (IQI) | 154.00 (146.00; 162.00) | 151.00 (146.00; 156.00) | 152.00 (146.00; 158.00) | **<0.001 [c]** |
| Weight | 1398 (100) | 1390 (100) | 2788 (100) | |
| Mean (SD) | 46.50 (13.62) | 43.46 (10.25) | 44.99 (12.15) | **<0.001 [b]** |
| Median (IQI) | 45.10 (36.80; 53.90) | 42.20 (36.40; 49.20) | 43.30 (36.60; 51.70) | **<0.001 [c]** |
| BMI-for-age | 1398 (100) | 1390 (100) | 2788 (100) | |
| Mean (SD) | 19.44 (4.30) | 19.00 (3.60) | 19.22 (3.97) | **<0.001 [b]** |
| Median (IQI) | 18.52 (16.25; 21.94) | 18.39 (16.41; 20.94) | 18.46 (16.31; 21.40) | 0.07 [c] |
| Body fat percentage | 1379 (100) | 1389 (100) | 2786 (100) | |
| Mean (SD) | 17.62 (11.65) | 23.75 (8.82) | 20.68 (10.78) | **<0.001 [b]** |
| Median (IQI) | 14.40 (8.10; 24.60) | 22.80 (17.30; 29.00) | 19.40 (12.30; 27.60) | **<0.001 [c]** |
| Mother's BMI | 1259 (100) | 1284 (100) | 2543 (100) | |
| Mean (SD) | 21.89 (2.85) | 22.01 (2.71) | 21.95 (2.78) | 0.28 [b] |
| Median (IQI) | 21.64 (20.09; 23.31) | 21.64 (20.31; 23.34) | 21.64 (20.20; 23.33) | 0.15 [c] |
| Normal, n (%) | 900 (71.5) | 911 (71.0) | 1811 (71.2) | 0.95 [a] |
| Overweight, n (%) | 226 (18.0) | 234 (18.2) | 460 (18.1) | |
| Obese, n (%) | 133 (10.6) | 139 (10.8) | 272 (10.7) | |
| Father's BMI | 1225 (100) | 1211 (100) | 2436 (100) | |
| Mean (SD) | 22.86 (2.84) | 22.60 (2.63) | 22.73 (2.74) | **0.02 [b]** |
| Median (IQI) | 22.86 (20.96; 24.44) | 22.51 (20.76; 24.22) | 22.68 (20.81; 24.22) | **0.01 [c]** |
| Normal, n (%) | 636 (51.9) | 683 (56.4) | 1319 (54.1) | **0.05 [a]** |
| Overweight, n (%) | 359 (29.3) | 338 (27.9) | 697 (28.6) | |
| Obese, n (%) | 230 (18.8) | 190 (15.7) | 420 (17.2) | |
| **Behavioral characteristics** | | | | |
| Transportation to school | 1393 (100) | 1386 (100) | 2779 (100) | |
| Walking, n (%) | 241 (17.3) | 212 (15.3) | 453 (16.3) | **0.02 [a]** |
| Bicycle, n (%) | 864 (62.0) | 829 (59.8) | 1693 (60.9) | |
| Parental assistance, n (%) | 198 (14.2) | 216 (15.6) | 414 (14.9) | |
| Other, n (%) | 90 (6.5) | 129 (9.3) | 219 (7.9) | |

(*Continued*)

**Table 1.** (Continued)

| | Boys (n = 1398) | Girls (n = 1390) | Total (n = 2788) | p-value |
|---|---|---|---|---|
| Do at least one type of sport | 1387 (100) | 1373 (100) | 2760 (100) | |
| No, n (%) | 66 (4.8) | 184 (13.4) | 250 (9.1) | **<0.001** [a] |
| Yes, n (%) | 1321 (95.2) | 1189 (86.6) | 2510 (90.9) | |
| Type of sport | 1321 (100) | 1189 (100) | 2510 (100) | |
| Badminton, n (%) | 129 (9.8) | 422 (35.5) | 551 (22.0) | **<0.001** [a] |
| Soccer/Football, n (%) | 821 (62.1) | 85 (7.1) | 906 (36.1) | |
| Shuttlecock kicking, n (%) | 166 (12.6) | 263 (22.1) | 429 (17.1) | |
| Swimming, n (%) | 114 (8.6) | 115 (9.7) | 229 (9.1) | |
| Others, n (%) | 91 (6.9) | 304 (25.6) | 395 (15.7) | |
| Physical mins per day | 1304 (100) | 1172 (100) | 2476 (100) | |
| Mean (SD) | 62.87 (36.37) | 45.28 (30.14) | 54.54 (34.69) | **<0.001** [b] |
| Median (IQI) | 60.00 (30.00; 60.00) | 30.00 (30.00; 60.00) | 50.00 (30.00; 60.00) | **<0.001** [c] |
| Play video games per day | 1392 (100) | 1384 (100) | 2776 (100) | |
| No gaming, n (%) | 467 (33.5) | 811 (58.6) | 1278 (46.0) | **<0.001** [a] |
| <1hour, n (%) | 293 (21.0) | 239 (17.3) | 532 (19.2) | |
| 1 hour, n (%) | 356 (25.6) | 197 (14.2) | 553 (19.9) | |
| 2 hours, n (%) | 199 (14.3) | 96 (6.9) | 295 (10.6) | |
| 3 hours, n (%) | 50 (3.6) | 26 (1.9) | 76 (2.7) | |
| ≥4 hours, n (%) | 27 (1.9) | 15 (1.1) | 42 (1.5) | |
| Watching TV/smartphone per day | 1386 (100) | 1378 (100) | 2764 (100) | |
| No watching, n (%) | 81 (5.8) | 67 (4.9) | 148 (5.4) | 0.13 [a] |
| <1hour, n (%) | 352 (25.4) | 364 (26.4) | 716 (25.9) | |
| 1 hour, n (%) | 508 (36.7) | 515 (37.4) | 1023 (37.0) | |
| 2 hours, n (%) | 345 (24.9) | 303 (22.0) | 648 (23.4) | |
| 3 hours, n (%) | 68 (4.9) | 81 (5.9) | 149 (5.4) | |
| ≥4 hours, n (%) | 32 (2.3) | 48 (3.5) | 80 (2.9) | |

Bold-faced p-values indicate statistical significance (p < 0.05)

†Sample size for individual characteristics may not equal total due to missing values

[#] equivalent to 23–25 kg/m$^2$ for the overweight group and ≥ 25 kg/m$^2$ for obesity group

[a] p-value is obtained from Pearson's chi-square test

[b] p-value was obtained from two sample t-test

[c] p-value was obtained from Wilcoxon rank-sum test

between the two definitions. Approximately 8.6% of all participants were defined as obese by the WHO Z-score criteria, of which the proportion in boys was triple than that in girls (12.9% vs. 4.2%). On the other hand, while using the IOTF reference, only 5.4% participated students were classified as obese, and the proportion in boys was two-fold higher than that in girls (7.6% vs. 3.2%).

Key differences in the classification of child's sex and age-specific overweight and obesity in accordance to different diagnostic criteria are shown in a series of graphs (Fig 1). Overall, the patterns of BMI-for-age classification tend to be similar between the IOTF and CDC cut-points, in contrast to the WHO Z-score for all ages and both sexes. For classifications in boys, the proportion of children defined as overweight or obese were greater in the younger age group (11 and 12 years old), then decreased with age. In addition, the differences between these classifications among boys for both the prevalence of overweight and obesity were

**Table 2. Prevalence of overweight and obesity of children by age and sex according to WHO Z-score and IOTF references (n = 2788).**

| Groups | Boys | | | Girls | | | Total | | |
|---|---|---|---|---|---|---|---|---|---|
| | n | % | 95% CI | n | % | 95% CI | n | % | 95% CI |
| **BMI-for-age group by Z-score [a]** | | | | | | | | | |
| **Overweight** | **271** | **19.4** | **[16.7–22.4]** | **213** | **15.3** | **[12.5–18.6]** | **484** | **17.4** | **[15.1–19.9]** |
| 11 years old | 75 | 20.7 | [15.9–26.5] | 55 | 15.6 | [11.4–21.0] | 130 | 18.2 | [14.6–22.5] |
| 12 years old | 79 | 22.7 | [18.7–27.3] | 53 | 14.8 | [10.9–19.7] | 132 | 18.7 | [15.5–22.3] |
| 13 years old | 72 | 19.4 | [15.1–24.6] | 57 | 15.9 | [11.7–21.3] | 129 | 17.7 | [14.3–21.7] |
| 14 years old | 45 | 14.2 | [10.7–18.6] | 48 | 15 | [11.2–19.7] | 93 | 14.6 | [11.9–17.8] |
| **Obese** | **181** | **12.9** | **[9.9–16.8]** | **59** | **4.2** | **[3.1–5.7]** | **240** | **8.6** | **[6.7–11.0]** |
| 11 years old | 64 | 17.7 | [13.2–23.3] | 19 | 5.4 | [3.4–8.5] | 83 | 11.6 | [8.9–15.0] |
| 12 years old | 62 | 17.8 | [13.2–23.6] | 15 | 4.2 | [2.4–7.2] | 77 | 10.9 | [8.2–14.3] |
| 13 years old | 25 | 6.7 | [4.1–10.9] | 12 | 3.4 | [1.8–6.2] | 37 | 5.1 | [3.4–7.6] |
| 14 years old | 30 | 9.5 | [5.7–15.2] | 13 | 4 | [2.2–7.4] | 43 | 6.7 | [4.4–10.2] |
| **BMI group by IOTF [b]** | | | | | | | | | |
| **Overweight** | **278** | **19.9** | **[17.0–23.2]** | **200** | **14.4** | **[11.5–17.9]** | **478** | **17.1** | **[14.7–19.9]** |
| 11 years old | 84 | 23.2 | [18.4–28.9] | 53 | 15.1 | [11.0–20.3] | 137 | 19.2 | [15.7–23.2] |
| 12 years old | 88 | 25.3 | [20.6–30.6] | 52 | 14.5 | [10.3–20.1] | 140 | 19.8 | [16.1–24.1] |
| 13 years old | 61 | 16.4 | [12.4–21.4] | 54 | 15.1 | [10.9–20.5] | 115 | 15.8 | [12.4–19.8] |
| 14 years old | 45 | 14.2 | [10.5–19.0] | 41 | 12.8 | [9.2–17.4] | 86 | 13.5 | [10.7–16.9] |
| **Obese** | **106** | **7.6** | **[5.4–10.6]** | **45** | **3.2** | **[2.3–4.5]** | **151** | **5.4** | **[4.0–7.3]** |
| 11 years old | 33 | 9.1 | [5.9–13.7] | 13 | 3.7 | [2.2–6.1] | 46 | 6.4 | [4.5–9.2] |
| 12 years old | 30 | 8.6 | [5.9–12.5] | 10 | 2.8 | [1.3–5.9] | 40 | 5.7 | [3.8–8.3] |
| 13 years old | 19 | 5.1 | [2.9–9.0] | 7 | 2 | [1.0–4.0] | 26 | 3.6 | [2.2–5.7] |
| 14 years old | 24 | 7.6 | [4.4–12.8] | 15 | 4.7 | [2.7–8.0] | 39 | 6.1 | [3.9–9.4] |

[a] equivalent to 1 SD-2 SD for the overweight group and $\geq$ 2 SD for obesity group

[b] equivalent to 25–30 kg/m$^2$ for the overweight group and $\geq$ 30 kg/m$^2$ for obesity group at the age of 18

notably substantial. In the obese category for boys, the WHO Z-score based prevalence was approximately 10–15% higher than the IOTF and CDC-based prevalence. In contrast, the patterns for classification in both the overweight and obesity categories appeared to be stable across all ages among girls, with the exception for the age of 14. The overweight and obesity prevalence by age, sex, and four different classifications are described in further detail in the S1 Table.

**Table 3** illustrates both the demographic and behavioral characteristics of participants stratified by BMI-for-age groups by using the WHO Z-score cut-off. While 7.7% of participants were categorized as thin, the prevalence of overweight and obesity within the sample population was 17.4% and 8.6%, respectively. The stunting prevalences were 6.99% in overall (6.08% among boys and 7.91% among girls) (S3 and S4 Tables). Obesity status was more frequently observed in those at younger age, male children, or those who lived in the South of Vietnam (p < 0.001 for all comparisons). Conversely, children from a rural area, living in the Center of Vietnam, or those belonging to the ethnic minority were more likely to be thin (p < 0.001 for all comparisons). Children with a high BMI were found to reside with parents, who also had a higher BMI as compared to others (p < 0.001 for all comparisons). In terms of behavioral factors, overweight and obese children generally spent more time playing video games and watching TV, as compared to those in the normal and thin category (p < 0.001). Interestingly, although the overwhelming percentage of children played at least one type of sport (nearly 90%), those who were classified as overweight or obese self-reported a greater time playing

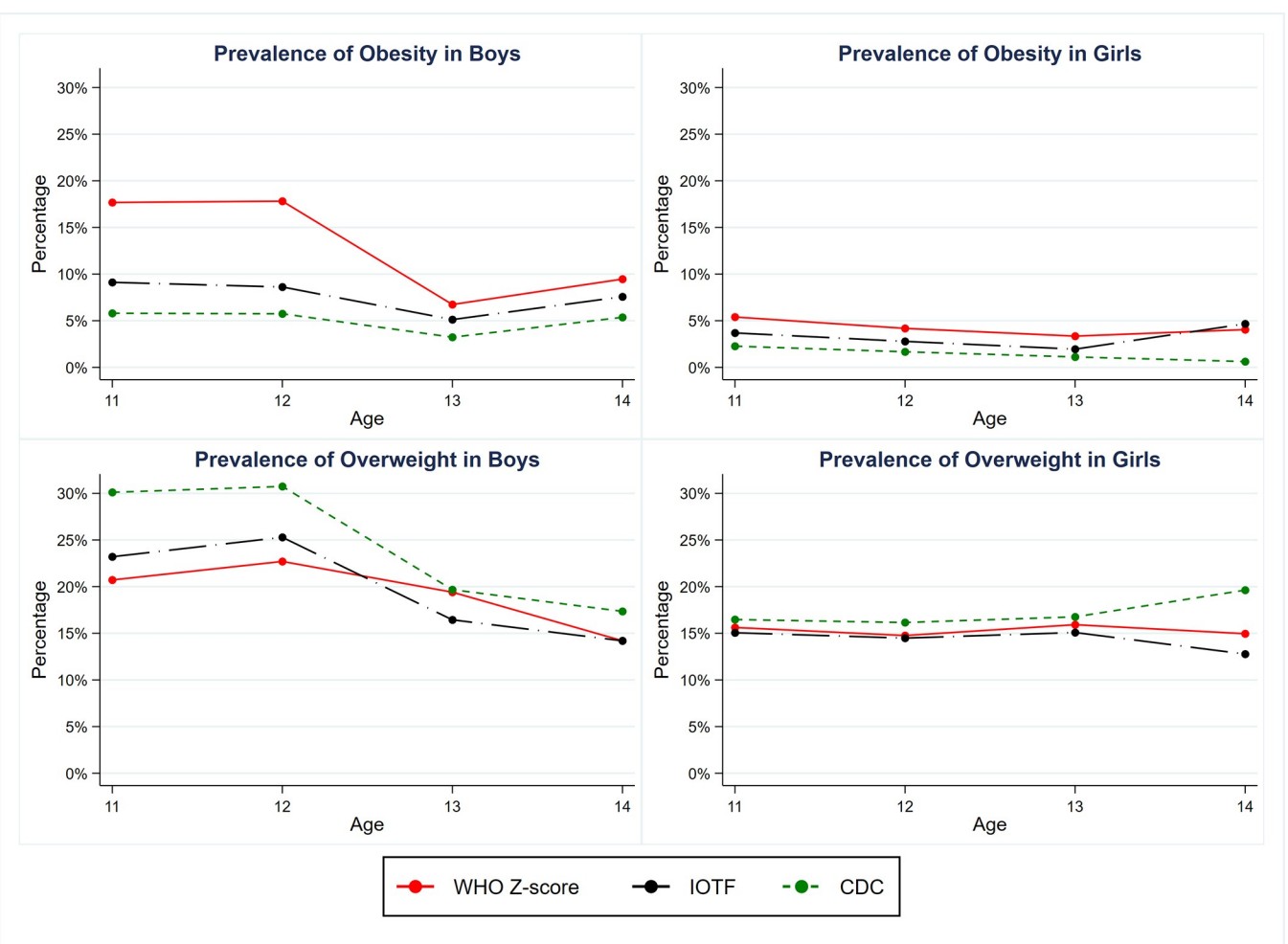

**Fig 1. Comparison of the prevalence of overweight and obese children by sex based on four criteria: WHO Z-score, IOTF and CDC reference.** Red solid line: WHO Z-score (equivalent to 1SD-2SD for the overweight group and ≥3SD for the obese group. Black dash-dot line: IOTF (equivalent to 25–30 kg/m$^2$ for the overweight group and ≥30 kg/m$^2$ for the obese group at the age of 18. Green short-dash line: CDC (defines overweight as 85$^{th}$ = BMI <95 percentiles, and obesity as BMI = 95$^{th}$ percentile).

sport as compared to normal and thin children (58.6 minutes and 66.3 minutes vs. 52.4 and 49.7 minutes, respectively, p < 0.001).

The sex-specific regression analysis of obesity alone, and a combinatory analysis of both overweight and obesity prevalence using the WHO Z-score cut-off was described in **Figs 2** and **3**. In regards to the obesity models, having an older age (PR = 0.75, 95%CI: 0.65–0.85) was associated with obesity among young Vietnamese boys. The prevalence of obesity among both sexes was shown to increase proportionately with their parents' BMI. In reference to the overweight/obesity combined models, we identified that an older age (PR = 0.83, 95%CI: 0.76–0.90) and belonging to any ethnic minority groups (PR = 0.43, 95%CI: 0.24–0.76) were also significant risk factors among boys. Consistently, parental BMI was also positively associated with an increased prevalence of overweight and obesity among both boys and girls. We also conducted additional sensitivity analyses to investigate if the exclusion of thin children from the presented models would affect the associations. Moreover, we found that without the thin children group in the model, the previously mentioned significant factors remained the same, we did not identify any meaningful differences.

**Table 3. Demographic and behavioral characteristics by BMI-for-age groups using WHO Z-core criteria.**

| BMI-for-age group | Thin | Normal | Overweight | Obesity | p-value |
|---|---|---|---|---|---|
| n(%) | **216 (7.7)** | **1848 (66.3)** | **484 (17.4)** | **240 (8.6)** | |
| Age group | 216 (100) | 1848 (100) | 484 (100) | 240 (100) | |
| 11 years old, n (%) | 68 (31.5) | 433 (23.4) | 130 (26.9) | 83 (34.6) | <**0.001** [a] |
| 12 years old, n (%) | 48 (22.2) | 450 (24.4) | 132 (27.3) | 77 (32.1) | |
| 13 years old, n (%) | 51 (23.6) | 512 (27.7) | 129 (26.7) | 37 (15.4) | |
| 14 years old, n (%) | 49 (22.7) | 453 (24.5) | 93 (19.2) | 43 (17.9) | |
| Sex | 216 (100) | 1848 (100) | 484 (100) | 240 (100) | |
| Boys, n (%) | 121 (56.0) | 825 (44.6) | 271 (56.0) | 181 (75.4) | <**0.001** [a] |
| Girls, n (%) | 95 (44.0) | 1023 (55.4) | 213 (44.0) | 59 (24.6) | |
| Region | 216 (100) | 1848 (100) | 484 (100) | 240 (100) | |
| North, n (%) | 60 (27.8) | 668 (36.1) | 149 (30.8) | 50 (20.8) | <**0.001** [a] |
| Center, n (%) | 96 (44.4) | 635 (34.4) | 140 (28.9) | 57 (23.8) | |
| South, n (%) | 60 (27.8) | 545 (29.5) | 195 (40.3) | 133 (55.4) | |
| Residency status | 216 (100) | 1848 (100) | 484 (100) | 240 (100) | |
| Urban, n (%) | 63 (29.2) | 711 (38.5) | 239 (49.4) | 129 (53.8) | <**0.001** [a] |
| Rural, n (%) | 153 (70.8) | 1137 (61.5) | 245 (50.6) | 111 (46.3) | |
| Ethnicity | 216 (100) | 1845 (100) | 483 (100) | 240 (100) | |
| Kinh, n (%) | 169 (78.2) | 1504 (81.5) | 449 (93.0) | 219 (91.3) | <**0.001** [a] |
| Other, n (%) | 47 (21.8) | 341 (18.5) | 34 (7.0) | 21 (8.8) | |
| Mother's BMI | 193 (100) | 1679 (100) | 446 (100) | 226 (100) | |
| Mean (SD) | 21.35 (2.51) | 21.77 (2.78) | 22.49 (3.29) | 22.96 (3.19) | <**0.001** [b] |
| Median (IQI) | 20.83 (19.74; 22.49) | 21.56 (20.00; 23.11) | 22.06 (20.81; 23.73) | 22.53 (20.83; 24.56) | <**0.001** [c] |
| Normal, n (%) | 154 (79.8) | 1229 (73.2) | 298 (66.8) | 130 (57.5) | <**0.001** [a] |
| Overweight, n (%) | 24 (12.4) | 294 (17.5) | 91 (20.4) | 51 (22.6) | |
| Obese, n (%) | 15 (7.8) | 156 (9.3) | 57 (12.8) | 45 (19.9) | |
| Father's BMI | 183 (100) | 1607 (100) | 428 (100) | 220 (100) | |
| Mean (SD) | 22.25 (4.36) | 22.53 (2.78) | 23.32 (2.52) | 23.80 (3.24) | <**0.001** [b] |
| Median (IQI) | 22.04 (19.95; 23.66) | 22.50 (20.76; 24.22) | 23.51 (21.71; 24.80) | 23.71 (22.04; 25.10) | <**0.001** [c] |
| Normal, n (%) | 121 (66.1) | 924 (57.5) | 184 (43.0) | 90 (40.9) | <**0.001** [a] |
| Overweight, n (%) | 41 (22.4) | 433 (26.9) | 150 (35.0) | 73 (33.2) | |
| Obese, n (%) | 21 (11.5) | 250 (15.6) | 94 (22.0) | 57 (25.9) | |
| Do at least a type of sport | 213 (100) | 1826 (100) | 481 (100) | 240 (100) | |
| No, n (%) | 16 (7.5) | 171 (9.4) | 46 (9.6) | 17 (7.1) | 0.55 [a] |
| Yes, n (%) | 197 (92.5) | 1655 (90.6) | 435 (90.4) | 223 (92.9) | |
| Physical mins per day, mean (SD) | 49.74 (27.47) | 52.43 (33.86) | 58.61 (34.33) | 66.28 (43.04) | <**0.001** [b] |
| Play video games per day | 214 (100) | 1839 (100) | 483 (100) | 240 (100) | |
| No gaming, n (%) | 115 (53.7) | 887 (48.2) | 203 (42.0) | 73 (30.4) | <**0.001** [a] |
| < = 1hour, n (%) | 44 (20.6) | 342 (18.6) | 93 (19.3) | 53 (22.1) | |
| 1 hour, n (%) | 28 (13.1) | 362 (19.7) | 105 (21.7) | 58 (24.2) | |
| 2 hours, n (%) | 18 (8.4) | 171 (9.3) | 66 (13.7) | 40 (16.7) | |
| 3 hours, n (%) | 6 (2.8) | 47 (2.6) | 12 (2.5) | 11 (4.6) | |
| ≥4 hours, n (%) | 3 (1.4) | 30 (1.6) | 4 (0.8) | 5 (2.1) | |
| Watching TV per day | 212(100) | 1833 (100) | 480 (100) | 239 (100) | |

(*Continued*)

**Table 3.** (Continued)

| BMI-for-age group | Thin | Normal | Overweight | Obesity | p-value |
|---|---|---|---|---|---|
| n(%) | **216 (7.7)** | **1848 (66.3)** | **484 (17.4)** | **240 (8.6)** | |
| No watching, n (%) | 17 (8.0) | 103 (5.6) | 18 (3.8) | 10 (4.2) | 0.18 [a] |
| < = 1hour, n (%) | 48 (22.6) | 472 (25.8) | 137 (28.5) | 59 (24.7) | |
| 1 hour, n (%) | 85 (40.1) | 678 (37.0) | 176 (36.7) | 84 (35.1) | |
| 2 hours, n (%) | 52 (24.5) | 432 (23.6) | 105 (21.9) | 59 (24.7) | |
| 3 hours, n (%) | 6 (2.8) | 102 (5.6) | 24 (5.0) | 17 (7.1) | |
| ≥4 hours, n (%) | 4 (1.9) | 46 (2.5) | 20 (4.2) | 10 (4.2) | |

Bold p-value indicates statistical significance (p < 0.05)

†Sample size for individual characteristics may not equal total due to missing values

[*] Children are defined as thinness if their weight-to-height was less than (-2SD), and normal if their weight-to-height was between (-2SD) and 1SD.

[#] equivalent to 23–25 kg/m$^2$ for the overweight group and ≥ 25 kg/m$^2$ for obesity

[a] p-value is obtained from Pearson's chi-square test

[b] p-value was obtained from one-way ANOVA test

[c] p-value was obtained from Kruskal-Wallis test

## Discussion

Our primary objective was to estimate the prevalence of overweight and obese children between the age 11 and 14 in Vietnam. While the overweight prevalence was one in every six children based on both the WHO Z-score and IOTF reference, the obesity prevalence fluctuated between 5.4% (using IOTF criteria) and 8.6% (using WHO Z-score criteria). When employing different reference classifications, the proportions of overweight and obesity were consistently higher among boys, in comparison to girls in the majority of all age groups. Regarding significant risk factors, parental BMI was associated with a higher likelihood of having overweight or obese children, regardless of sex. Additionally, older age and belong to ethnic minority group were associated with an increased prevalence of childhood overweight and obesity among boys only.

### Prevalence of overweight and obesity

Although most of the earliest studies which surveyed the prevalence of overweight and obesity among Vietnamese children was conducted in Ho Chi Minh City–a metropolitan city of Vietnam, numerous papers indicate different findings [12–20]. The details describing the various findings of previous studies is shown in the **S2 Table**. During 2007–2010, it was evident that there was a low proportion of overweight (around 10%) and obese (less than 2%) children according to IOTF reference [13, 16] and WHO Z-score [12] were reported while malnutrition of Vietnamese children also remained a significant issues at that time [12]. However, the status of overweight and obesity among Vietnamese children aged 11–14 has increased swiftly in the last decade; this prevalence as defined by the IOTF has oscillated between 12.5% and 28.3% from 2013 to 2016 [14, 18, 19]. Surprisingly, a recent study conducted by To et al. in 2018 has indicated that the prevalence of overweight and obesity among urban Vietnamese children was more than 50% when using the WHO Z-score classification [20]. Conversely, another study in 2011 utilized the WHO Z-score scale and reported a low prevalence, which was described as less than 10% among 11-to-14-year-old children in Bac Giang–a small and rural province in the north of Vietnam [15]. In our sample, the combined prevalence of overweight and obesity was 32.2% for the urban area and 21.6% for the rural area (using the WHO Z-score cut-off). The discrepancy between the two regions can be explained by the rapid changes

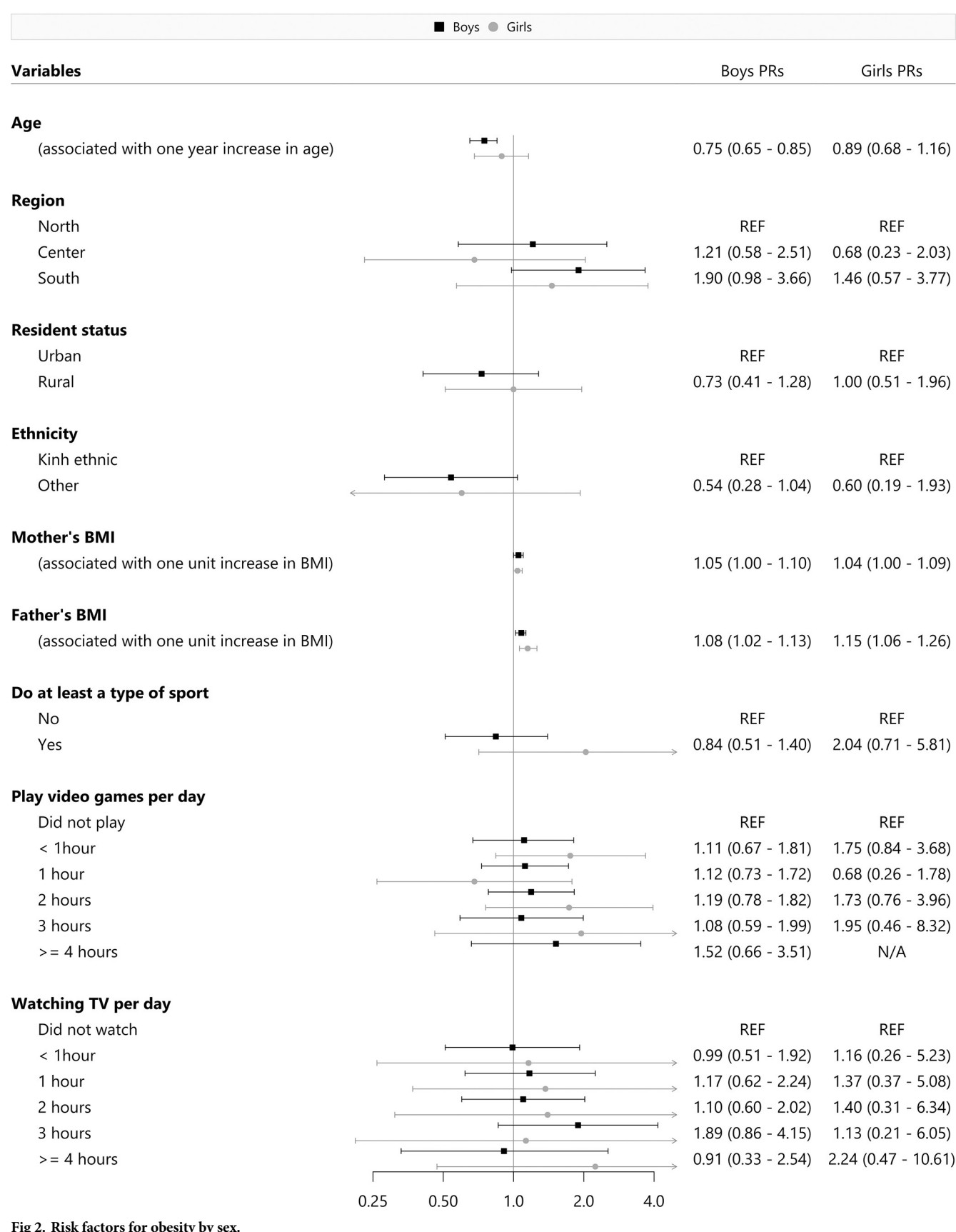

| Variables | Boys PRs | Girls PRs |
|---|---|---|
| **Age** | | |
| (associated with one year increase in age) | 0.75 (0.65 - 0.85) | 0.89 (0.68 - 1.16) |
| **Region** | | |
| North | REF | REF |
| Center | 1.21 (0.58 - 2.51) | 0.68 (0.23 - 2.03) |
| South | 1.90 (0.98 - 3.66) | 1.46 (0.57 - 3.77) |
| **Resident status** | | |
| Urban | REF | REF |
| Rural | 0.73 (0.41 - 1.28) | 1.00 (0.51 - 1.96) |
| **Ethnicity** | | |
| Kinh ethnic | REF | REF |
| Other | 0.54 (0.28 - 1.04) | 0.60 (0.19 - 1.93) |
| **Mother's BMI** | | |
| (associated with one unit increase in BMI) | 1.05 (1.00 - 1.10) | 1.04 (1.00 - 1.09) |
| **Father's BMI** | | |
| (associated with one unit increase in BMI) | 1.08 (1.02 - 1.13) | 1.15 (1.06 - 1.26) |
| **Do at least a type of sport** | | |
| No | REF | REF |
| Yes | 0.84 (0.51 - 1.40) | 2.04 (0.71 - 5.81) |
| **Play video games per day** | | |
| Did not play | REF | REF |
| < 1hour | 1.11 (0.67 - 1.81) | 1.75 (0.84 - 3.68) |
| 1 hour | 1.12 (0.73 - 1.72) | 0.68 (0.26 - 1.78) |
| 2 hours | 1.19 (0.78 - 1.82) | 1.73 (0.76 - 3.96) |
| 3 hours | 1.08 (0.59 - 1.99) | 1.95 (0.46 - 8.32) |
| >= 4 hours | 1.52 (0.66 - 3.51) | N/A |
| **Watching TV per day** | | |
| Did not watch | REF | REF |
| < 1hour | 0.99 (0.51 - 1.92) | 1.16 (0.26 - 5.23) |
| 1 hour | 1.17 (0.62 - 2.24) | 1.37 (0.37 - 5.08) |
| 2 hours | 1.10 (0.60 - 2.02) | 1.40 (0.31 - 6.34) |
| 3 hours | 1.89 (0.86 - 4.15) | 1.13 (0.21 - 6.05) |
| >= 4 hours | 0.91 (0.33 - 2.54) | 2.24 (0.47 - 10.61) |

**Fig 2. Risk factors for obesity by sex.**

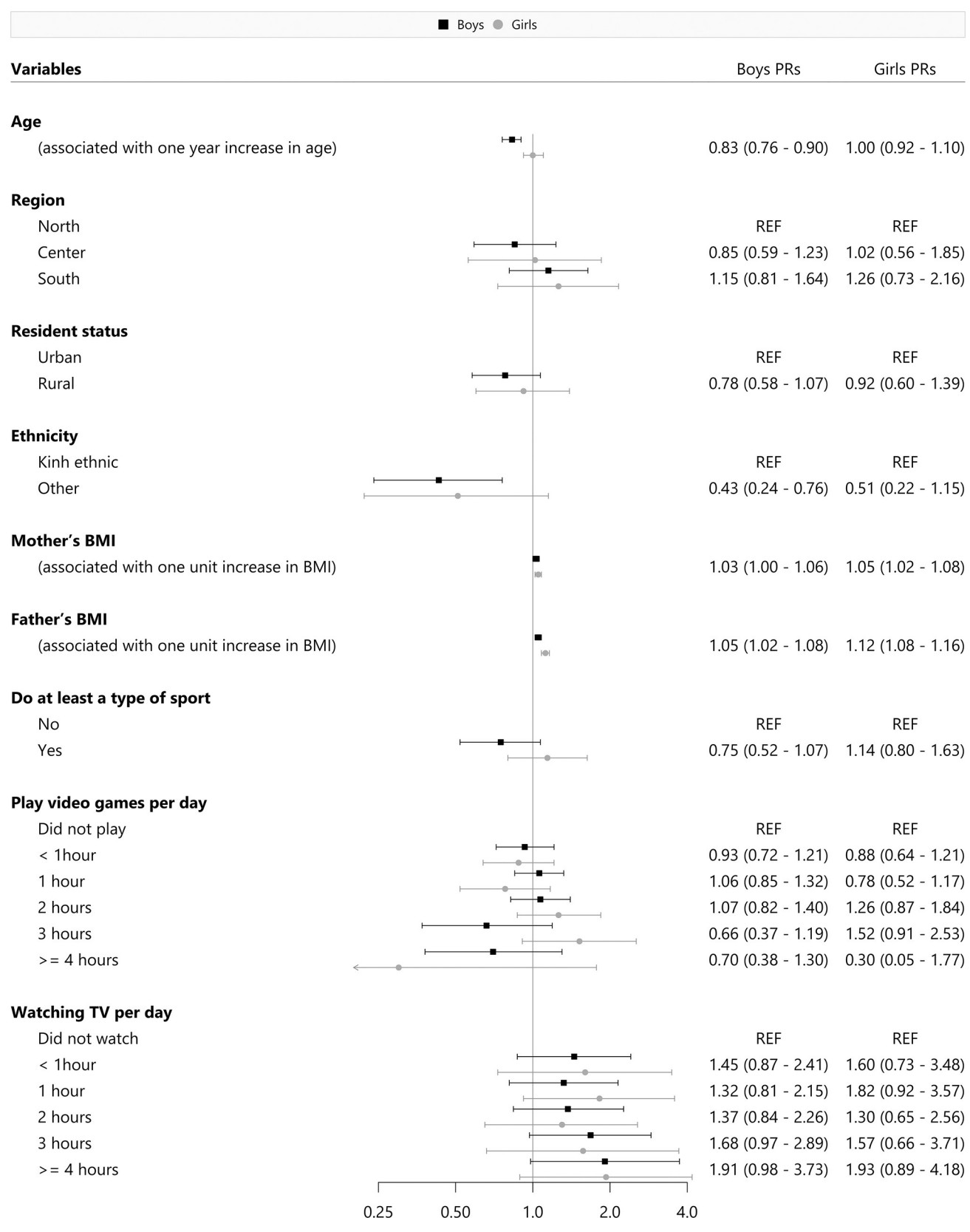

| Variables | Boys PRs | Girls PRs |
|---|---|---|
| **Age** | | |
| (associated with one year increase in age) | 0.83 (0.76 - 0.90) | 1.00 (0.92 - 1.10) |
| **Region** | | |
| North | REF | REF |
| Center | 0.85 (0.59 - 1.23) | 1.02 (0.56 - 1.85) |
| South | 1.15 (0.81 - 1.64) | 1.26 (0.73 - 2.16) |
| **Resident status** | | |
| Urban | REF | REF |
| Rural | 0.78 (0.58 - 1.07) | 0.92 (0.60 - 1.39) |
| **Ethnicity** | | |
| Kinh ethnic | REF | REF |
| Other | 0.43 (0.24 - 0.76) | 0.51 (0.22 - 1.15) |
| **Mother's BMI** | | |
| (associated with one unit increase in BMI) | 1.03 (1.00 - 1.06) | 1.05 (1.02 - 1.08) |
| **Father's BMI** | | |
| (associated with one unit increase in BMI) | 1.05 (1.02 - 1.08) | 1.12 (1.08 - 1.16) |
| **Do at least a type of sport** | | |
| No | REF | REF |
| Yes | 0.75 (0.52 - 1.07) | 1.14 (0.80 - 1.63) |
| **Play video games per day** | | |
| Did not play | REF | REF |
| < 1hour | 0.93 (0.72 - 1.21) | 0.88 (0.64 - 1.21) |
| 1 hour | 1.06 (0.85 - 1.32) | 0.78 (0.52 - 1.17) |
| 2 hours | 1.07 (0.82 - 1.40) | 1.26 (0.87 - 1.84) |
| 3 hours | 0.66 (0.37 - 1.19) | 1.52 (0.91 - 2.53) |
| >= 4 hours | 0.70 (0.38 - 1.30) | 0.30 (0.05 - 1.77) |
| **Watching TV per day** | | |
| Did not watch | REF | REF |
| < 1hour | 1.45 (0.87 - 2.41) | 1.60 (0.73 - 3.48) |
| 1 hour | 1.32 (0.81 - 2.15) | 1.82 (0.92 - 3.57) |
| 2 hours | 1.37 (0.84 - 2.26) | 1.30 (0.65 - 2.56) |
| 3 hours | 1.68 (0.97 - 2.89) | 1.57 (0.66 - 3.71) |
| >= 4 hours | 1.91 (0.98 - 3.73) | 1.93 (0.89 - 4.18) |

**Fig 3. Risk factors for overweight and obesity by sex.**

of lifestyle among urban children, progression towards the so-called Western diets, an increased consumption of sugar-sweetened beverages, and a decrease in physical activities. These factors and a variety of other all contribute to a higher risk for the onset of overweight and obesity in youth, as compared to children living in rural areas [12–16]. After adjusting for other demographic and behavioral factors, we also identified the prevalence of overweight and obese children between urban and rural areas of Vietnam is not statistically significant. Moreover, the sampling conducted in previous studies had a miniscule sample size which was not representative of the entire country [12–20]. This stark decision to assess the Vietnamese population via different methods is a major factor which may influence the observed differences in the status of childhood overweight and obesity prevalence between our study and previous surveys.

Among Asian countries, the prevalence of overweight and obesity also varies considerably. For example, the prevalence may range from 26.2% to over 40% in India, Korea, and China [31, 32, 47, 48]. A systematic review among Asian countries also reported an childhood overweight prevalence of 15% and 10% for obesity, which was in line with the findings of our present study [5]. The inconsistent results across previous studies in Vietnam and the aforementioned countries may also be due to substantial difference in the diagnostic criteria for overweight/obese children, age ranges, and the time periods when the studies were conducted [5, 31, 32]. In addition, discrepancies of socioeconomic status are present across countries could play an essential role in the underlying burden of obesity; these social and financial factors directly impact both eating habits and leisure-time physical activities.

### Reference populations and diagnostic criteria

As mentioned above, we found several differences among obesity and overweight prevalence when using different criteria/reference populations, our observations are consistent with a variety of surveys all around the world [5, 31, 32, 34–37]. Among the references used in our study, the IOTF (which combined surveys from the Netherlands, United Kingdom, Singapore, Hong Kong, Brazil, and the United States from 1963 to 1993) and WHO Z-score and percentile (which is derived from 4 separate dataset in the United State between 1963–1974) were widely utilized for international comparisons [28–30]. Meanwhile, the cut-off point developed by the CDC in 2000 was limited to the United State and occasionally, applied towards the assessment of the Canadian and Australian population [30]. As a result, the prevalence of overweight and obesity for children may vary when using different definitions (such as BMI is a quantification of body fat or weight); there is presently a lack of reference scales that are fully representative and specific to the diversity of Asian populations. Therefore, reporting prevalence by the application of different criteria is crucial for the time being. Numerous studies in Asia, including India [31], South Korea [32] and China [33, 38], have already adopted this reporting strategy. Nevertheless, all of the references in our study diagnose overweight and obesity by using BMI. This approach is both simple and practical but it has a variety of limitations. The most considerable limitation is that the application of BMI could not differentiate between fat and muscle tissues density, which varies significantly by sex, age, and body composition [30]. Several methods have been adopted to measure fat tissues such as hydrodensitometry, dual-energy x-ray absorptiometry (DEXA/DXA), and computerized tomography (CT) [30]. However, compared to BMI, many of these measurements require a significant investment into complex and expensive equipment, which limits their utilization in community-based studies.

### Age

We identified that the prevalence of overweight and obesity decreased with age among boys, this still existed after adjusting for potential confounders. Previous studies have also reported

that the prevalence of obesity was significantly higher among younger ages in both developing and developed countries [13, 34, 49]. The reasons supporting this phenomenon are still presently unknown, but we suspect that the possible cause could be hormonal changes during the period one transition through puberty. These physiological alterations may lead to dramatic changes in height and weight, particularly in boys. During this period, Asian adolescents may follow a different height and weight growth curve as compared to the Caucasian counterparts, which can potentially lead to the drastic difference in BMI that we observed [49].

## Gender

Sex differences which drive unique prevalences of overweight and obesity in individuals is noteworthy, especially for the male group which has a significantly higher prevalence of both conditions as compared to their female counterparts. A similar observation was reported in all of published studies in regards to youth prevalence of overweight and obesity in Vietnam [12–14, 17–20], in addition to other populations in China [21, 33, 39, 47], India [31], South Korea [32], Portugal [34, 50], and Asian American [49]. These difference may be due to the sex differences which influence individual body composition, hormone biology, the onset and duration of puberty, physical activities patterns, and dietary habits [51]. For example, young female children may focus on their appearance in the eyes of others as compared to their male counterparts, and therefore may attempt to reduce or limit their dietary portions [52, 53]. As a result, young females may feel more concerned about their weight and attempt to control food intake to a "normal range" as compared to young males. Additionally, previous Vietnamese findings have reported that while adolescent females completed more housework and homework, boys are more likely to be attracted to computer games which leads to an increase in their sedentary time [19].

## Parental BMI

Earlier studies have demonstrated a positive association between childhood overweight/obesity status and parental BMI [12, 13, 21, 23, 40, 54]. Our findings agree with these findings as the increase of BMI of maternal or paternal was associated with an increased prevalence of being overweight and obese in both boys and girls. On the other hand, a significant highlight of our study is the observation that parents of thin children had a lower mean BMI as compared to those with overweight and obese children. Therefore, the results suggest that parents' eating behavior and their perception of weight possibly influence their children, particularly at the pre-adolescent stage when all members of families share meals in Vietnam as well as in other developing nations. For the prevention and control of overweight and obesity in children, programs in those nations should consider an approach directed towards the whole family to bring about sustainable effects, rather than the individual.

## Physical activities

Physical inactivity and increments of sedentary time were found to be important predictors of childhood obesity globally [13, 15, 50]. In our study, we identified that the more children watched television, the more likely to be overweight or obese, even it was not significant. One surprising result of our study was that both overweight and obese children reported a greater amount of time allocated to recreational sports, this is in comparison to those who have an average or below average BMI. This observation may be due to either self-reported response bias or the overestimation of physical activities in overweight and obese children, which should be taken into account in the future studies on children' perception of physical activities and diets with an objective measurement of energy input and output.

### Strengths and limitations

It is the first study to estimate the national overweight and obesity prevalence of Vietnamese children aged 11–14 using both WHO 2007 and IOTF criteria. With a relatively large sample size and employment of a clustered random sampling method, our findings could be potentially generalized to all children of the same age across Vietnam. Additionally, weight and height in this study was directly measured by a team of healthcare professionals who have a wide range of experience in collecting health information at the time of the interview, this factor could mitigate the potential bias of recall information.

However, this study has some limitations. First and foremost, our sample is limited to only children in secondary school within the age range of 11 to 14, rather than all adolescent ages. The school selection in each cluster was not chosen by randomly, thus it could also result in some sampling bias. Second, nutrition behaviors and physical activities, particularly the input and output energy measurement, which are strongly associated with overweight and obesity prevalence were not well documented in this study. Third, the questionnaire was not validated in Vietnamese children, so it could be a threat to internal validity of our findings. Finally, a cross-sectional study design is not able to determine a causal relationship, so the risk factors identified in this study can only indicate an association.

## Conclusion

This study has distinguished an inflated national prevalence of overweight and obesity in children between the age 11 to 14 in Vietnam. The findings from this study indicate the alarming health burden of childhood overweight and obesity in the country. Targeting both overweight and obesity through novel interventions should apply a greater focus on children who are males, have overweight or obese parents, and live in the southern regions of Vietnam.

## Supporting information

**S1 Fig. A flow chart of sampling method.**
(PDF)

**S1 Table. Overweight and obesity prevalence by age and sex based on four BMI metrics including WHO Z-score, WHO percentile, IOTF and CDC references.**
(PDF)

**S2 Table. Summary of previous findings in Vietnamese children aged 11–16 years old.**
(PDF)

**S3 Table. Children' height by age and sex.**
(PDF)

**S4 Table. Prevalence of stunting and thinness by age and sex.**
(PDF)

**S1 File. Questionnaire (Vietnamese and English version).**
(PDF)

## Acknowledgments

PHD and HVM conceived the study design and protocol. NDT and DTV carried out data collection and data management. NTNP, BPL and PTT analyzed the data and developed the

paper structure. All authors were involved in writing the paper and final approval of the submitted and published versions.

The authors are grateful to all local guides, interviewers and colleagues for their logistic support. We are thankful for 2880 students and their parent for approval, spending their precious time and sharing their information with us. We thank Jonathan Josephs-Spaulding all English editing support.

## Author Contributions

**Conceptualization:** Huong Duong Phan, Thi Ngoc Phuong Nguyen, Phuong Linh Bui, Thanh Tung Pham.

**Formal analysis:** Thi Ngoc Phuong Nguyen, Phuong Linh Bui, Thanh Tung Pham.

**Funding acquisition:** Huong Duong Phan, Hoang Van Minh.

**Investigation:** Huong Duong Phan, Tuan Vu Doan, Duc Thanh Nguyen.

**Methodology:** Huong Duong Phan, Tuan Vu Doan, Duc Thanh Nguyen, Hoang Van Minh.

**Project administration:** Huong Duong Phan, Tuan Vu Doan, Duc Thanh Nguyen, Hoang Van Minh.

**Resources:** Tuan Vu Doan, Hoang Van Minh.

**Software:** Thi Ngoc Phuong Nguyen.

**Supervision:** Hoang Van Minh.

**Visualization:** Thi Ngoc Phuong Nguyen, Phuong Linh Bui, Thanh Tung Pham.

**Writing – original draft:** Thi Ngoc Phuong Nguyen, Phuong Linh Bui.

**Writing – review & editing:** Huong Duong Phan, Thi Ngoc Phuong Nguyen, Phuong Linh Bui, Thanh Tung Pham, Tuan Vu Doan, Duc Thanh Nguyen, Hoang Van Minh.

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
