## [Decision Letter · Decision Letter 0]

9 Jul 2020

PONE-D-20-09289

Overweight and obesity among Vietnamese school-aged children: National prevalence estimates based on the World Health Organization and International Obesity Task Force definition

PLOS ONE

Dear Dr. Bui,

Thank you for submitting your manuscript to PLOS ONE. After careful consideration, we feel that it has merit but does not fully meet PLOS ONE’s publication criteria as it currently stands. Therefore, we invite you to submit a revised version of the manuscript that addresses the points raised during the review process.

I would also like to apologize for the delay in the peer review process for reasons beyond our control; it was very difficult to get reviewers during these difficult times. As you will see that now you have received some valuable reviewer comments.

We look forward to receiving your revised manuscript.

Kind regards,

Madhavi Bhargava, MD

Academic Editor

PLOS ONE

Journal Requirements:

2. Please provide additional details regarding participant consent. In the ethics statement in the Methods and online submission information, please ensure that you have specified (1) whether consent was informed and (2) what type you obtained (for instance, written or verbal). If your study included minors, state whether you obtained consent from parents or guardians. If the need for consent was waived by the ethics committee, please include this information.”

3. Please include additional information regarding the survey or questionnaire used in the study and ensure that you have provided sufficient details that others could replicate the analyses. For instance, if you developed a questionnaire as part of this study and it is not under a copyright more restrictive than CC-BY, please include a copy, in both the original language and English, as Supporting Information. Moreover, please include more details on how the questionnaire was pre-tested, and whether it was validated.

Reviewers' comments:

Reviewer's Responses to Questions

**Comments to the Author**

1. Is the manuscript technically sound, and do the data support the conclusions?

Reviewer #1: Partly

Reviewer #2: Partly

2. Has the statistical analysis been performed appropriately and rigorously? 

Reviewer #1: No

Reviewer #2: Yes

3. Have the authors made all data underlying the findings in their manuscript fully available?

Reviewer #1: No

Reviewer #2: Yes

4. Is the manuscript presented in an intelligible fashion and written in standard English?

Reviewer #1: Yes

Reviewer #2: Yes

5. Review Comments to the Author

Reviewer #1: General comments

1. The paper has useful data to assess the prevalence of overweight and obesity in Vietnam and helps us to understand the disease burden; however there is a need for better presentation and analysis of the data.

Specific comments

1. Line 108-The multi stage sampling need to explain, why schools were not randomly selected. It would be useful to know how many schools were present in each cluster on average. It is mentioned that schools with highest number of children were selected when there were more than one school. However, this could result in bias of sampling as schools with more children could also be accommodating children from low socio economic groups and vice versa. A flow chart of number of schools and children excluded would be helpful to understand.

2. Line 126-What were the socio economic factors that were collected as it is unclear from the result section and whether they represented the population. Also was onset of menarche was noted.

3. Line 132-No mention on how body fat has been calculated but was shown in tables in methods section apart from height and weight.

4. Line 182- Instead of BMI, BAZ scores may be presented as BMI is not useful in this age group.

5. Line 195- Table 1, it would be better to present column wise proportions instead of rowwise proportions, as the p values compare the differences in boys and girls.

6. Height is a strong predictor of lean mass and therefore obesity, it would be useful to include them in the models. Further, future studies can help in assessing secular trends over the year using a similar design, which would help us to assess the extent of reduction of stunting. Also report the prevalence of stunting in this population using WHO growth standards for understanding the existence of dual burden of malnutrition.

7. Line 248- Table 3, present column wise proportions as explained above.

8. Figure 2- It is important to have models adjusted for age and sex to be adjusted to have clear explanation of the other variables as risk factors. The discussion needs to be written based on these changes.

Reviewer #2: Thank you for the opportunity to review this article. The work is interesting, but some aspects should be taken into account before publication.

Comments and suggestions for Authors:

- The aim of the study should be more clearly defined in the introduction

- Keywords: It needs to be sorted in alphabetical order. No capital letters at the beginning of the word.

- In the methodology section there is no information about the body composition measurements, but in the Table 1 there is a "Body fat percentage". Please add a description of the methodology to this section.

- Please do not start the sentence with numbers (line 174).

- In the Table 1 there is an information about mother’s and father’s BMI. Was the measurement taken or did the parents give their declared body weight and height in the questionnaire? Please add this information to the methodology section.

- Did the survey participants complete the paper-based questionnaire alone or with their parents?

- There is information about obtaining the consent of the Bioethics Committee. Please provide the number of this consent.

6. PLOS authors have the option to publish the peer review history of their article (what does this mean?). If published, this will include your full peer review and any attached files.

Reviewer #1: No

Reviewer #2: No

---

## [Author Response · Author response to Decision Letter 0]

8 Sep 2020

Dear Sir/Madam, 

We would like to express our sincere thanks to the editor for your comments on our manuscript. We hope that a revised version of the manuscript will be considered for publication. 

We have modified the paper in response to the insightful comments of the editor and reviewers. Furthermore, we have rewritten sections of the manuscript and hope that this complies with the reviewers’ remarks. 

We have responded to the comments point by point. Our answers to the reviewers’ questions were highlighted in yellow. All the line numbers mentioned were based on the revised manuscript.

****

Journal Requirements:

Thank you for your comment. We revised our manuscript accordingly.

2. Please provide additional details regarding participant consent. In the ethics statement in the Methods and online submission information, please ensure that you have specified (1) whether consent was informed and (2) what type you obtained (for instance, written or verbal). If your study included minors, state whether you obtained consent from parents or guardians. If the need for consent was waived by the ethics committee, please include this information.”

Thank you. Study participants in our study were children (minors) so according to the local IRB requirement, we obtained written informed consent from parents of those children. The detailed information leaflet and written informed consent were sent to parents by school teachers before the interview day. After that, those teachers collected all written inform consents from children’s parents, and then local health professionals who were trained as data collectors by National Hospital of Endocrinology obtained verbal agreement from these children before starting any step of the interview process. We addded the information to line 120-123.

3. Please include additional information regarding the survey or questionnaire used in the study and ensure that you have provided sufficient details that others could replicate the analyses. For instance, if you developed a questionnaire as part of this study and it is not under a copyright more restrictive than CC-BY, please include a copy, in both the original language and English, as Supporting Information. Moreover, please include more details on how the questionnaire was pre-tested, and whether it was validated.

Thank you for your comments. We have added the Vietnamese and English version of the questionnaire to the supplementary section for the readers to refer to. (Supplement S4). 

We added “The questionnaire was designed based on the previous studies in Vietnam among adult population and the doctoral thesis of Tran et al on children aged 6-14 in Hanoi, Vietnam (26,27). The questionnaire was not validated in Vietnamese children but pretested for grammatical errors and readability among 30 children in Ha Long city, Quang Ninh province, Vietnam before the full-scale data collection. All the questions were revised and finalized by a panel of health experts from National Hospital of Endocrinology and Hanoi University of Public Health.” in line 125-131. 

The questionnaire is not validated in Vietnamese children so it could be a threat to internal validity of our results. Therefore, we acknowledge this limitation in the discussion section (line 415-416).

Thank you for your comments. There is an ethical restriction on sharing a de-identified dataset due to the ethical requirement of local IRB, namely Institutional Science Review Board of the National Hospital of Endocrinology. If anyone interested in our data, they may directly contact our research team via email (ntnp@huph.edu.vn) or the IRB member (PhD. MD. Le Quang Toan, Vice Chairman of the IRB of National Hospital of Endocrinology via letoan.endo@gmail.com or (+84) 904007097).

Reviewer #1

1. Line 108-The multi stage sampling need to explain, why schools were not randomly selected. It would be useful to know how many schools were present in each cluster on average. It is mentioned that schools with highest number of children were selected when there were more than one school. However, this could result in bias of sampling as schools with more children could also be accommodating children from low socio economic groups and vice versa. A flow chart of number of schools and children excluded would be helpful to understand.

Thank you for pointing this out, we have added an acknowledgment of this limitation in line 412-413: “The school selection in each cluster was not chosen by randomly, thus it could also result in some sampling bias.” 

Commonly, in each cluster in rural area (ward/commune/town), there was only 1 secondary school (50% clusters located in rural area). In urban area, there were commonly 1-3 secondary schools in each cluster. We assumed that the most crowded school in each cluster represented the whole cluster better than the other schools and the expert panel made a consensus about this during the implementation. We acknowledged that this step would weaken the representation of the sample. The best solution should be randomly choosing a secondary school among all eligible schools in each cluster instead of choosing the most crowded school. This is a limitation of this study and we acknowledged it in the manuscript. Thank you for your valuable comments. 

As your recommendation to further explain the survey design, we provided a flow chart as your recommendation in Supplementary S1. We also included it here for your reference.

2. Line 126-What were the socio economic factors that were collected as it is unclear from the result section and whether they represented the population. Also was onset of menarche was noted.

Thank you. We added more detail on socio-economic factors explanation in line 135-137: “Our paper-based questionnaire included a variety of questions on several socioeconomic factors (including age, ethnicity, region, resident type), and general health status (such as physical activities and lifestyle behavior) (Supplementary S4).” We also included the questionnaire in the supplemental section (S4) for readers to refer to. For the onset of menarche, we did not collect this information so that we couldn’t present in our result section.

3. Line 132-No mention on how body fat has been calculated but was shown in tables in methods section apart from height and weight.

Thanks for your comments. The body fat percentage was directly extracted from the TANITA digital scale when we measured students’ weight. We added this information in line 135-136: “We further directly collected information about children’s body fat percentage though this scale.”

4. Line 182- Instead of BMI, BAZ scores may be presented as BMI is not useful in this age group.

Thank you for pointing this out. We used the WHO package to measure the overweight and obesity prevalence, thus the BMI score in our paper was BAZ score, as WHO’s guideline. We made change accordingly in line 196.

5. Line 195- Table 1, it would be better to present column wise proportions instead of rowwise proportions, as the p values compare the differences in boys and girls.

Thank you. We have presented the data in column wise proportions. For example:

 Boys (n=1398) Girls (n=1390) Total (n = 2788)

Demographic characteristics 

Age group 1398 (100) 1390 (100) 2788 (100) 

 11 years old, n (%) 362 (25.9) 352 (25.3) 714 (25.6)

 12 years old, n (%) 348 (24.9) 359 (25.8) 707 (25.4)

 13 years old, n (%) 371 (26.5) 358 (25.8) 729 (26.1)

 14 years old, n (%) 317 (22.7) 321 (23.1) 638 (22.9)

6. Height is a strong predictor of lean mass and therefore obesity, it would be useful to include them in the models. Further, future studies can help in assessing secular trends over the year using a similar design, which would help us to assess the extent of reduction of stunting. Also report the prevalence of stunting in this population using WHO growth standards for understanding the existence of dual burden of malnutrition.

Thank you for your comments. We reported the stunning prevalence in Table 3 which were 7.7% children in total (of which, 56% were boys and 44% were girls). 

We believe that height should be considered as a mediator between children’s age and overweight/obesity in this case rather than a confounder; therefore, we believe that height should not be included in the model in this case.

Based on prior knowledge from the literature, we created a simple causal diagram (also known as DAG - directed acyclic graph) to examine the relationship between potential factors and obesity. Hernán et al. and VanderWeele et al. showed that such an approach would be less bias compared to traditional approaches, such as backward selection and forward selection procedures.

REF:

https://academic.oup.com/aje/article/155/2/176/108106

https://www.ncbi.nlm.nih.gov/pmc/articles/PMC6447501/

https://pubmed.ncbi.nlm.nih.gov/21627630/?dopt=Abstract

In this approach, it is recommended to adjust for confounder, not the mediator to preserve the association between the main exposure and the main outcome.

Height should be on the causal pathway from age to obesity (i.e height is the mediator in this relationship). In children, as age increases, height and weight commonly increase; and obesity/overweight is a downstream of high BMI (i.e. weight adjusted for height). Therefore, adjusting for both age and height in this model with the obesity/overweight outcome may cause collinearity and could distort the association between age (and probably other variables) and the main outcome (obesity/overweight). 

As age is an important exposure that must be included in this model, adding height into the model at the same time in this case does not give us more useful information but distorting associations that we want to see.

7. Line 248- Table 3, present column wise proportions as explained above.

Thank you. We have presented the data in column wise proportions. 

8. Figure 2- It is important to have models adjusted for age and sex to be adjusted to have clear explanation of the other variables as risk factors. The discussion needs to be written based on these changes.

Thank you for your comment. We used two separate models for boys and girls in this study because we thought sex should be considered as an effect modifier rather than a confounder to be included in the model. 

According to Vander Weele, “We say that there is effect modification in distribution across strata of Q for the effect of A on Y if P(Ya|Q = q) varies with q.”

(https://www.ncbi.nlm.nih.gov/pmc/articles/PMC4249691/)

The sex-specific effects due to different biological mechanism (https://www.annualreviews.org/doi/abs/10.1146/annurev-nutr-071816-064827) and different contextual effects of sex on obesity (https://academic.oup.com/advances/article/3/4/491/4591492) have been reported in previous studies.

Knol and VanderWeele et al. also recommended presenting data within stratum of effect modifier (in this case, a model for boys and a model for girls) and the interaction terms in simple model (one binary exposure, one binary outcome, and one interaction)

(https://www.ncbi.nlm.nih.gov/pmc/articles/PMC3324457/).

Moreover, Lovejoy et al. recommended that data should be presented separately for each gender (https://pubmed.ncbi.nlm.nih.gov/19021872/).

In our model, we did find that the association of age, parental BMI, ethnicity, playing sport, watching TV with obesity/overweight did vary with different types of sex. If we add sex into the model, it will require adding several interaction terms of sex with those variables (including both categorical and continuous variables), which makes the model too complicated and difficult to interpret. Therefore, we decided to implement two models separately for males and females.

Reviewer #2: 

- The aim of the study should be more clearly defined in the introduction

Thank you for your comments. We re-write this sentence as follows: “In this study, we aim to conduct to estimate the national prevalence of overweight and obesity in children, by using both the WHO and IOTF criteria to identify associated factors which influence these conditions among school-aged children in Vietnam” (line 78-81).

- Keywords: It needs to be sorted in alphabetical order. No capital letters at the beginning of the word.

Thank you. We made changes and re-order it as follows: children; childhood obesity; low- and middle-income countries; overweight (line 36).

- In the methodology section there is no information about the body composition measurements, but in the Table 1 there is a "Body fat percentage". Please add a description of the methodology to this section.

Thank you for pointing this out. The body fat percentage was directly extracted from the TANITA digital scale when we measured students’ weight. We added this information (“We further directly collected information about children’s body fat percentage though this scale.”) in line 142-143.

- Please do not start the sentence with numbers (line 174).

Thank you. We rewrite it as follows: “There were 2788 children (96.81%) with completed information on age, sex, weight and height included in the analysis.” (line 186)

- In the Table 1 there is an information about mother’s and father’s BMI. Was the measurement taken or did the parents give their declared body weight and height in the questionnaire? Please add this information to the methodology section.

Thanks for your comments. The parents themselves gave us their information on body weight and height in another paper-based survey. We have included in it the method section as follows: “Parents’ weight and height were collected via an independent survey sent directly to them for self-report.” (line 143-145)

- Did the survey participants complete the paper-based questionnaire alone or with their parents?

Thank you for your question. The survey participants were interviewed by healthcare professionals who trained as data collectors. So, the data collector was the one who filled in the questionnaire based on the answer of study participants (as described in line 120-125). Meanwhile, another survey was directly sent to their parents to ask for parents’ information like weight, height (we added this information in line 143-145).

- There is information about obtaining the consent of the Bioethics Committee. Please provide the number of this consent.

Thank you for your comment. We added the number of this consent (No 1133/QD-BVNTTW) in line 89.

***

Thank you very much for your consideration. We are looking forward to your comments and suggestions.

Yours sincerely,

Linh Phuong Bui, MD, MPH

---

## [Decision Letter · Decision Letter 1]

28 Sep 2020

Overweight and obesity among Vietnamese school-aged children: National prevalence estimates based on the World Health Organization and International Obesity Task Force definition

PONE-D-20-09289R1

Dear Dr. Bui,

We’re pleased to inform you that your manuscript has been judged scientifically suitable for publication and will be formally accepted for publication once it meets all outstanding technical requirements.

Kind regards,

Madhavi Bhargava, MD

Academic Editor

PLOS ONE

Additional Editor Comments (optional):

Both the reviewers have commended the way all the reviewers comments have been addressed comprehensively. There is a small suggestion by the Reviewer - 1. You are free to take a call on what he has suggested. I too agree with him as far as mean heights are concerned. It will also improve the citability of your work since height is getting more and more attention for South East Asia region. But of course the acceptance of the article is not contingent on that.

Reviewers' comments:

Reviewer's Responses to Questions

**Comments to the Author**

1. If the authors have adequately addressed your comments raised in a previous round of review and you feel that this manuscript is now acceptable for publication, you may indicate that here to bypass the “Comments to the Author” section, enter your conflict of interest statement in the “Confidential to Editor” section, and submit your "Accept" recommendation.

Reviewer #1: All comments have been addressed

Reviewer #2: All comments have been addressed

2. Is the manuscript technically sound, and do the data support the conclusions?

Reviewer #1: Yes

Reviewer #2: Yes

3. Has the statistical analysis been performed appropriately and rigorously? 

Reviewer #1: Yes

Reviewer #2: Yes

4. Have the authors made all data underlying the findings in their manuscript fully available?

Reviewer #1: Yes

Reviewer #2: Yes

5. Is the manuscript presented in an intelligible fashion and written in standard English?

Reviewer #1: Yes

Reviewer #2: Yes

6. Review Comments to the Author

Reviewer #1: In table 1, Correct is as BMI as this is no BMI for age Z score.

I could not find any data on stunting in Table 3, but nevertheless this is fine with me. If it is possible to report mean height of the children, it would be helpful for researchers interested in secular trends in heights, as South East Asian children are one of the shortest in the world.

Reviewer #2: Thank you for the opportunity to review this resubmission. Authors have done a nice job addressing reviewers' comments. Thank you. I am ok with acceptance.

7. PLOS authors have the option to publish the peer review history of their article (what does this mean?). If published, this will include your full peer review and any attached files.

Reviewer #1: **Yes: **Dr. Raja Sriswan Mamidi

Reviewer #2: No

---

## [Editor Report · Acceptance letter]

2 Oct 2020

PONE-D-20-09289R1 

Overweight and obesity among Vietnamese school-aged children: National prevalence estimates based on the World Health Organization and International Obesity Task Force definition 

Dear Dr. Bui:

I'm pleased to inform you that your manuscript has been deemed suitable for publication in PLOS ONE. Congratulations! Your manuscript is now with our production department. 

Kind regards, 

on behalf of

Dr. Madhavi Bhargava 

Academic Editor

PLOS ONE